# A Fault Warning Method for Electric Vehicle Charging Process Based on Adaptive Deep Belief Network

Dexin Gao [1], Yi Wang [1], Xiaoyu Zheng [1] and Qing Yang [2,*]

1   College of Automation and Electronic Engineer, Qingdao University of Science & Technology, Qingdao 266061, China; gaodexin@qust.edu.cn (D.G.); 4019040038@mails.qust.edu.cn (Y.W.); 4020040050@mails.qust.edu.cn (X.Z.)

2   College of Information Science and Technology, Qingdao University of Science & Technology, Qingdao 266061, China

*   Correspondence: 03390@qust.edu.cn; Tel.: +86-187-5326-0566

**Abstract:** If an accident occurs during charging of an electric vehicle (EV), it will cause serious damage to the car, the person and the charging facility. Therefore, this paper proposes a fault warning method for an EV charging process based on an adaptive deep belief network (ADBN). The method uses Nesterov-accelerated adaptive moment estimation (NAdam) to optimize the training process of a deep belief network (DBN), and uses the historical data of EV charging to construct the ADBN of the normal charging process of an EV model. The real-time data of EV charging is obtained and input into the constructed ADBN model to predict the output, calculate the Pearson coefficient between the predicted output and the actual measured value, and judge whether there is a fault according to the size of the Pearson coefficient to realize the fault warning of the EV charging process. The experimental results show that the method is not only able to accurately warn of a fault in the EV charging process, but also has higher warning accuracy compared with the back propagation neural network (BPNN) and conventional DBN methods.

**Keywords:** charging process of EV; fault warning; deep belief network; NAdam; Pearson coefficient

## 1. Introduction

Under the dual pressure of the energy crisis and environmental degradation, EVs, as the main development direction of new energy vehicles, have attracted increasing attention [1,2]. With the continuous development of the EV industry, the safety and reliability of charging has become the focus of attention. Various types of power batteries are the power source of EVs. Once the power battery fails in the process of charging, it is likely to lead to a fire accident, thus causing serious economic losses. Therefore, in the process of charging an EV, it is especially important to be able to effectively predict the occurrence of power battery failures, provide early warning before they occur, and take measures to deal with them.

Battery fault diagnosis methods can be divided into model-based methods and non-model-based methods [3,4]. Zhang, et al. [5] propose a method for the monitoring and warning of EV charging faults based on a battery model is proposed to judge whether the charging process is normal by comparing the charging response information simulated by the battery model with the battery charging status information. The method is a model-based method, which requires the establishment of an accurate electrochemical, electrothermal or other type of model depending on the type of battery. Tran, et al. [6] established the equivalent circuit model of a lithium-ion battery that depends on the state of charge, temperature and state of health, which has a high accuracy and can be effectively monitored for lithium batteries. In [7], the performance of three different equivalent circuit models was studied and compared using the chemical composition of four kinds of lithium battery, and the best model for each lithium battery was determined. Model-based methods

are required for signal processing and a large amount of calculation when dealing with complex and non-linear systems, which makes them difficult to apply to a variety of EVs, so the versatility is poor. The non-model-based method relies less on battery modeling, has stronger applicability, and has been widely used in the field of battery fault diagnosis. Reference [8] proposes a fault diagnosis and identification method for an EV based on BPNN, which can accurately identify charging fault levels and bring some assurance to EV charging safety.

The structure of BPNN is based on a shallow network model, which belongs to the traditional machine learning method. However, the EV charging process data contain rich information about EV charging under different working conditions, which has the characteristics of strong coupling and multidimensionality. This makes the traditional machine learning methods easy to fall into local optima and poor generalization ability during training, so it is difficult to realize the fault warning of an EV during charging. Deep learning, as a new field of neural networks in machine learning, has been widely applied to data mining, prediction and classification based on the working principle of human brain [9]. DBN is a typical data-driven deep learning model with outstanding ability to extract data features and handle high-dimensional as well as non-linear data [10], and has achieved good results in the field of fault warning [11–14]. Zheng, et al. [11] used an improved variational mode decomposition (VMD) and DBN method to achieve the failure warning of vulnerable parts of the wind turbine, and verify its feasibility and effectiveness. Li, et al. [12] adopted DBN to achieve a fault warning for inter-turn short circuits in the excitation winding of synchronous generators, with desired results. Chen, et al. [13] proposed a fault warning method for unmanned aerial vehicle (UAV) sensors based on DBN to achieve online fault detection of rotorcraft UAV flight system to realize the purpose of rapid warning. Huang, et al. [14] used DBN to construct an early warning model for landslide meteorology and realize the early warning of landslides triggered by rainfall. It can be seen that DBN has great advantages in the field of warning. However, the charging process fault warning of EV shows a trend of large data, and the traditional DBN will cause the problem of slow learning rate and a difficulty choosing the learning rate of the model when solving samples with high dimension, complex structure and large data volume, and when the adaptive ability is poor.

Aiming at the fault warning problem of the EV charging process, this paper proposes a fault warning method for the EV charging process. The method uses NAdam algorithm to optimize the training process of DBN, which can design adaptive learning rate for different parameters and realize the dynamic adjustment of learning rate in the model training process. As a deep learning method, the method deeply mines the historical data of the EV charging process, and constructs the adaptive deep belief network (ADBN) model of normal charging process. The real-time EV charging data is input into the ADBN model to predict the output, and the Pearson coefficient between the predicted output and the actual measured value is calculated. If the Pearson coefficient is less than the set threshold, the fault will be warned and the charging of EV will be cut off. The effectiveness and superiority of this paper's method are demonstrated by conducting warning experiments on charging voltage faults and charging current faults of an EV, and comparing them with other methods.

The rest of the paper is organized as follows. Section 2 briefly describes the principle of EV fault warning implementation. Section 3 introduces the principle of ADBN, which involves the structure and training process of DBN, NAdam algorithm, Pearson coefficient, and fault warning process. In Section 4, an experimental validation is performed to implement the fault warning for the charging process of an EV. Finally, the paper concludes in Section 5.

## 2. Problem Description

The idea of fault warning in an EV charging process is to obtain all physical quantities of all types of EV during charging through charging piles and store them in a database.

According to the unique code of an EV and using the historical charging data of this EV in the database, the deep learning network model belonging to this EV charging process is constructed. The real-time physical quantities of EV charging are obtained and input into the constructed exclusive deep network model to obtain the desired prediction data. The correlation between the predicted data and the actual measurements is observed to determine whether there is a fault in EV charging, thus realizing fault warning for the EV charging process. The fault warning of the EV charging process can effectively avoid charging safety accidents caused by high charging voltage, high current or high temperature. The charging process of the EV and its warning diagram are shown in Figure 1.

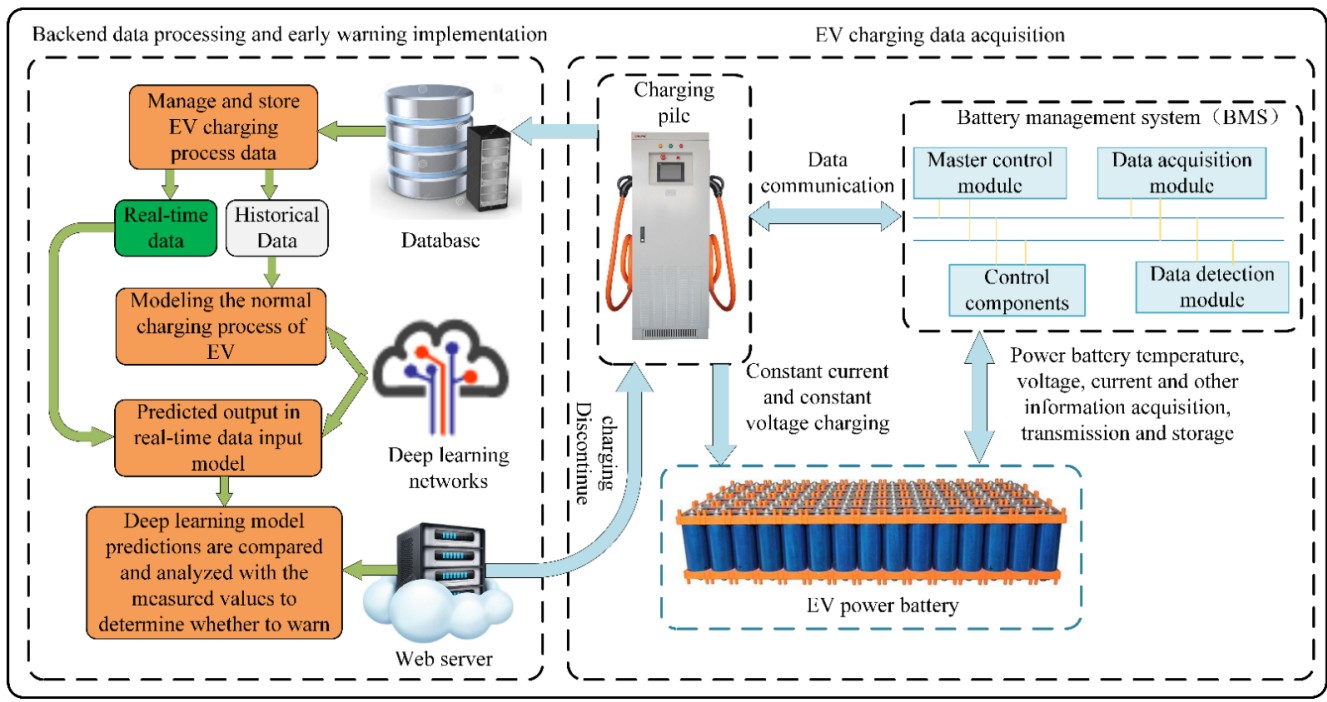

**Figure 1.** Electric vehicle (EV) charging process and its warning block diagram.

When the EV is charging, it will communicate with the charging pile. The battery management system (BMS) sends information such as the required voltage, current and temperature of the EV power battery to the charging pile, and the charging pile outputs energy to charge the EV according to the required voltage and current. The required voltage and current of EV will be changed in real time during charging, and the voltage and current output of the charging pile will be adjusted accordingly.

The data communication between the EV and charging piles complies with the Chinese national standard "GB/T 27,930 Communication Protocol between Non-vehicle Conductive Charger and Battery Management System for Electric Vehicles". According to this communication protocol, the physical quantities of the EV charging process that can be obtained are shown in Table 1.

**Table 1.** Physical quantities of EV charging process.

| Charging Process Physical Quantities | Unit | Precision | Period/ms |
|---|---|---|---|
| Rated capacity of power battery | Ah | 0.1 | 250 |
| Rated voltage of power battery | V | 0.1 | 250 |
| Maximum allowable individual voltage | V | 0.01 | 500 |
| Maximum allowable charging current | A | 0.1 | 500 |
| Power battery nominal energy | kW·h | 0.1 | 500 |
| Maximum allowable charging voltage | V | 0.1 | 500 |

**Table 1.** *Cont.*

| Charging Process Physical Quantities | Unit | Precision | Period/ms |
|---|---|---|---|
| Maximum allowable temperature | °C | 1 | 500 |
| Power battery initial state of charge (SOC) | % | 0.1 | 500 |
| Power battery initial voltage | V | 0.1 | 500 |
| Power battery required voltage | V | 0.1 | 50 |
| Power battery required current | A | 0.1 | 50 |
| Charging voltage measurement value | V | 0.1 | 250 |
| Charging current measurement value | A | 0.1 | 250 |
| Maximum individual voltage of power battery | V | 0.01 | 250 |
| Power battery current SOC | % | 1 | 250 |
| Maximum individual temperature of power battery | °C | 1 | 250 |

## 3. Electric Vehicle (EV) Charging Fault Warning Based on Adaptive Deep Belief Network (ADBN)

### 3.1. Structure and Training Process of DBN

A DBN is a deep neural network composed of s multiple restricted Boltzmann machine (RBM) and a BPNN stack [15], which uses an unsupervised greedy learning algorithm to adjust the connection weights of each RBM layer and a supervised learning approach to optimize the network parameters.

Figure 2 shows the structure of the DBN model and its training process for normal charging voltage of EV. As can be seen from Figure 2, each RBM consists of a visible layer $V_k = (v_1, v_2, \cdots, v_n)$ and a hidden layer $H_k = (h_1, h_2, \cdots, h_m)$. The visible layer $V_1$ and the hidden layer $H_1$ form RBM$_1$, the hidden layer $H_1$ as the visible layer of RBM$_2$ and the hidden layer $H_2$ form RBM$_2$, and so on. The lines in Figure 2 represent the weights between the connected neurons, $W_k = \{w_{i,j}\} \in R^{n \times m}$ is the connection weight between the visible layer and the hidden layer of the $k$th RBM, and $A_k = \{a_i\} = R^n$ and $B_k = \{b_j\} = R^m$ are the visible layer bias and hidden layer bias of the $k$th RBM. Thus, only three parameters are required to determine an RBM.

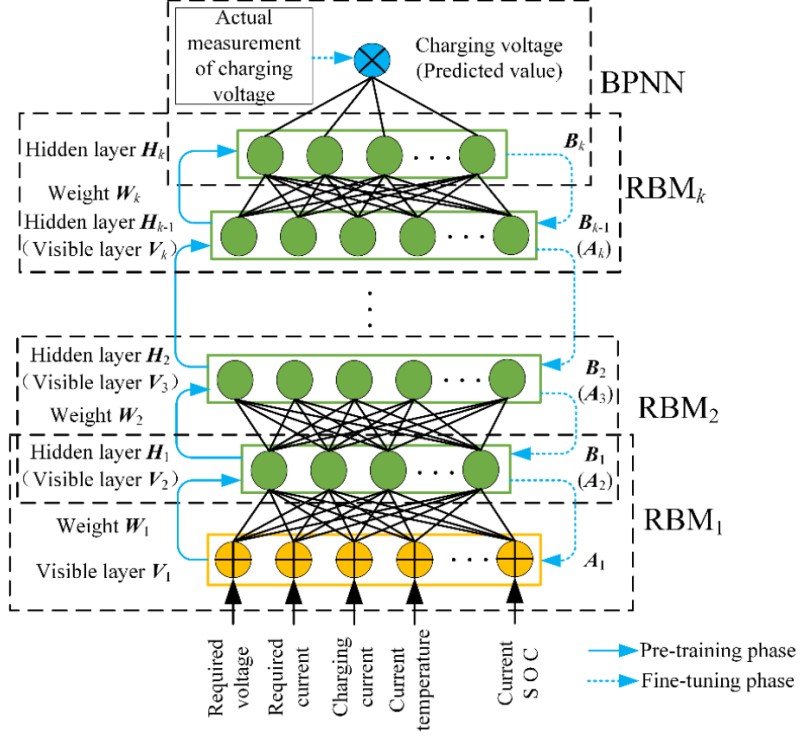

**Figure 2.** Structure and training diagram of deep belief network (DBN) model for normal charging voltage.

For the DBN model with normal charging voltage, the energy function of its internal RBM is defined as follows:

$$E(\boldsymbol{V}_k, \boldsymbol{H}_k \mid \boldsymbol{\theta}_k) = -\boldsymbol{A}_k^{\mathrm{T}} \boldsymbol{V}_k - \boldsymbol{B}_k^{\mathrm{T}} \boldsymbol{H}_k - \boldsymbol{V}_k^{\mathrm{T}} \boldsymbol{W}_k \boldsymbol{H}_k \tag{1}$$

where $\boldsymbol{V}_k$ and $\boldsymbol{H}_k$ denote the binary states of all units in the $k$th visible and hidden layer. The lower energy function indicates a more ideal state of the network, that is, the lower prediction error for the EV charging voltage. By regularizing and exponentializing the energy function, the joint probability distribution of the RBM can be obtained as follows:

$$P(\boldsymbol{V}_k, \boldsymbol{H}_k \mid \boldsymbol{\theta}_k) = \frac{\exp(-E(\boldsymbol{V}_k, \boldsymbol{H}_k \mid \boldsymbol{\theta}_k))}{Z(\boldsymbol{\theta}_k)} \tag{2}$$

$$Z(\boldsymbol{\theta}_k) = \sum\nolimits_{\boldsymbol{V}_k, \boldsymbol{H}_k} \exp(-E(\boldsymbol{V}_k, \boldsymbol{H}_k \mid \boldsymbol{\theta}_k)) \tag{3}$$

where $Z(\boldsymbol{\theta}_k)$ is the partition function, which represents the sum of all possible state energy functions of the set of $\boldsymbol{V}_k$ and $\boldsymbol{H}_k$ nodes in the normal charging voltage DBN model, and is used as the objective function of the optimization algorithm. According to the structural characteristics of the RBM, the probability that the $i$th unit $v_i$ of the visible layer $\boldsymbol{V}_k$ and the $j$th unit $h_j$ of the hidden layer $\boldsymbol{H}_k$ are activated can be expressed as follows:

$$P(v_i = 1 \mid \boldsymbol{H}_k) = \sigma(a_i + \sum_{i=1}^{m} h_j w_{ij}) \tag{4}$$

$$P(h_j = 1 \mid \boldsymbol{V}_k) = \sigma(b_j + \sum_{i=1}^{n} v_i w_{ij}) \tag{5}$$

where $\sigma(x) = 1/(1 + e^{-x})$ is the sigmoid activation function.

The DBN training process for normal charging voltage contains two stages of pre-training and fine-tuning. In the pre-training phase, $RBM_1$ receives information on the EV's required voltage, required current, charging current and temperature, and trains each RBM in a bottom-up sequence using a layer-by-layer greedy learning algorithm to achieve the extraction of high-level features of the input data and the update of the connection weights of the training network. The output data is the predicted charging voltage. In the fine-tuning phase, BPNN takes the predicted charging voltage as the input and the actually measured charging voltage as the output, and continuously adjusts and optimizes the network parameters from top to bottom in the way of supervised learning.

### 3.2. NAdam Algorithm

NAdam is the addition of Nesterov momentum to the adaptive moment estimation (Adam) [15–20]. Taking the DBN model of optimizing the normal charging voltage as an example, through literature [19], the update rules of Adam are obtained as follows:

$$\boldsymbol{g}_t = \nabla_{\boldsymbol{\theta}_t} J(\boldsymbol{\theta}_t) \tag{6}$$

$$\boldsymbol{m}_t = \beta_1 \boldsymbol{m}_{t-1} + (1 - \beta_1) \boldsymbol{g}_t \tag{7}$$

$$\boldsymbol{v}_t = \beta_2 \boldsymbol{v}_{t-1} + (1 - \beta_2) \boldsymbol{g}_t^2 \tag{8}$$

$$\hat{\boldsymbol{m}}_t = \frac{\boldsymbol{m}_t}{1 - \beta_1^t} \tag{9}$$

$$\hat{\boldsymbol{v}}_t = \frac{\boldsymbol{v}_t}{1 - \beta_2^t} \tag{10}$$

$$\boldsymbol{\theta}_{t+1} = \boldsymbol{\theta}_t - \frac{\eta}{\sqrt{\hat{\boldsymbol{v}}_t} + \varepsilon} \hat{\boldsymbol{m}}_t \tag{11}$$

where $\theta = \{\theta_1, \theta_2, \theta_3, \cdots \theta_k\}$ is the parameter of the normal charging voltage DBN model; $g_t$ is the gradient vector during normal charging voltage DBN model training; $\eta$ represents the learning rate of the normal charging voltage DBN model training; $J(\theta_t)$ is the partition function of the RBM in the normal charging voltage DBN model, that is, Equation (3); $\nabla_{\theta_t}$ is the partial derivative of $J(\theta_t)$ and $\theta$; $m_t$ and $v_t$ are the first-order moment (mean) and second-order moment (variance) of the gradient during the training of the normal charging voltage DBN model; $\hat{m}_t$ and $\hat{v}_t$ represent the deviation correction of $m_t$ and $v_t$, which are used to offset the deviation; $\beta_1$ and $\beta_2$ are the exponential decay rates of $m_t$ and $v_t$; $\varepsilon$ is the correction parameter to ensure that the denominator is non-zero; $t$ is the number of iterations in the training of the normal charging voltage DBN model. Bringing Equation (7) into Equations (9) and (11) yields:

$$\theta_{t+1} = \theta_t - \frac{\eta}{\sqrt{\hat{v}_t + \varepsilon}}\left(\frac{\beta_1 m_{t-1}}{1 - \beta_1^t} + \frac{(1 - \beta_1)g_t}{1 - \beta_1^t}\right) \tag{12}$$

The $m_{t-1}/1 - \beta_1^t$ in brackets is the deviation correction estimate of the momentum vector at the previous moment of the normal charging voltage DBN model, which can be obtained by replacing $\hat{m}_{t-1}$ with:

$$\theta_{t+1} = \theta_t - \frac{\eta}{\sqrt{\hat{v}_t + \varepsilon}}\left(\beta_1 \hat{m}_{t-1} + \frac{(1 - \beta_1)g_t}{1 - \beta_1^t}\right) \tag{13}$$

Now adding Nesterov momentum, the deviation correction estimate $\hat{m}_t$ of the current momentum vector of the normal charging voltage DBN model is directly used to replace the deviation-corrected estimate $\hat{m}_{t-1}$ of the previous momentum, which leads to the update rule of NAdam of the normal charging voltage DBN model as follows:

$$\theta_{t+1} = \theta_t - \frac{\eta}{\sqrt{\hat{v}_t + \varepsilon}}\left(\beta_1 \hat{m}_t + \frac{(1 - \beta_1)g_t}{1 - \beta_1^t}\right) \tag{14}$$

The traditional momentum algorithm has the disadvantage that the learning rate will not change in the training process and uses a single learning rate to update the weight [17]. However, NAdam designs independent adaptive learning rates for different parameters by calculating the first-order moments and second-order moments of the gradient, so that NAdam not only has stronger constraints on the learning rate, but also has a more direct impact on the update of the gradient. After the optimization of the NAdam algorithm, DBN can adapt to the corresponding charging data in the prediction of physical quantities in the normal charging process of different EVs, and has more powerful adaptability.

### 3.3. Pearson Coefficient

In this paper, Pearson's correlation coefficient is used as the discriminant condition for fault warning of the EV charging process, which is used to measure the degree of linear correlation between the predicted value of the model and actual measured values [21]. When the correlation between the predicted value and the actual measured value is low, it indicates that a fault may occur in this charging process, and its expression is:

$$r = \frac{\sum\limits_{i=1}^{n}(X_i - \overline{X})(Y_i - \overline{Y})}{\sqrt{\sum\limits_{i=1}^{n}(X_i - \overline{X})^2}\sqrt{\sum\limits_{i=1}^{n}(Y_i - \overline{Y})^2}} \tag{15}$$

where $n$ is the number of predicted value and actual measured value; $X_i$ and $Y_i$ are the observed value of $i$ point corresponding to predicted value $X$ and actual measured value $Y$ respectively; $\overline{X}$ and $\overline{Y}$ are the average number of $X$ and $Y$ samples respectively.

The value range of the Pearson coefficient is −1~1. When the value is 1, it indicates that there is a complete positive correlation between the predicted value and the actual measured value; When the value is 0.8~1, it indicates that there is an extreme correlation between the predicted value and the actual measured value. In order to ensure the accuracy of warning, this paper takes the Pearson coefficient as 0.8 as the warning limit.

*3.4. EV Charging Process Fault Warning Process*

Charging voltage fault, charging current fault, temperature fault and SOC fault may occur mainly during the charging process of an EV. Through Pearson correlation coefficient analysis, the fault types and fault discrimination methods are shown in Table 2. The fault warning of the EV charging process can be mainly divided into two stages: EV normal charging model training and real-time fault warning. Because the model training method and fault warning method are the same for each fault, we only take the fault warning of charging voltage and charging current as an example to realize the charging fault warning of EV. The warning process diagram of EV charging process is shown in Figure 3, and the specific implementation process is as follows:

1. Obtain the historical data of EV charging process, and divide it into normal charging data and fault charging data.
2. Data normalization of normal charging data and fault charging data.
3. Constructing the normal charging voltage model ADBN1 and the normal charging current model ADBN2 for EV by two stages of pre-training and fine-tuning using normal charging data.
4. Input the fault charging data into the constructed ADBN1 and ADBN2 models to predict the charging voltage and charging current, calculate the Pearson coefficient between the predicted charging voltage and charging current and the measured charging voltage and charging current, perform fault warning when the Pearson coefficient is less than the set expectation value. Calculate the ratio between the number of fault warnings and the actual number of faults to test the warning performance of the models.
5. The ADBN model that meets the requirements is applied to real-time charging fault warning for the EV.

**Table 2.** Fault types and identification methods table.

| Number | Fault Type | Fault Identification Method | Fault Description |
|---|---|---|---|
| 1 | Charging voltage fault | The Pearson coefficient between model prediction and actual measurement is less than 0.8 | Bias fault—charging voltage is higher or lower than the required voltage |
| 2 | Charging current fault | The Pearson coefficient between model prediction and actual measurement is less than 0.8 | Bias fault—charging current is higher or lower than the required current |
| 3 | Temperature fault | The Pearson coefficient between model prediction and actual measurement is less than 0.8 | Bias fault—the measured value of temperature is widely different from the predicted value |
| 4 | SOC fault | The Pearson coefficient between model prediction and actual measurement is less than 0.8 | Bias fault—the measured value of SOC is widely different from the predicted value |

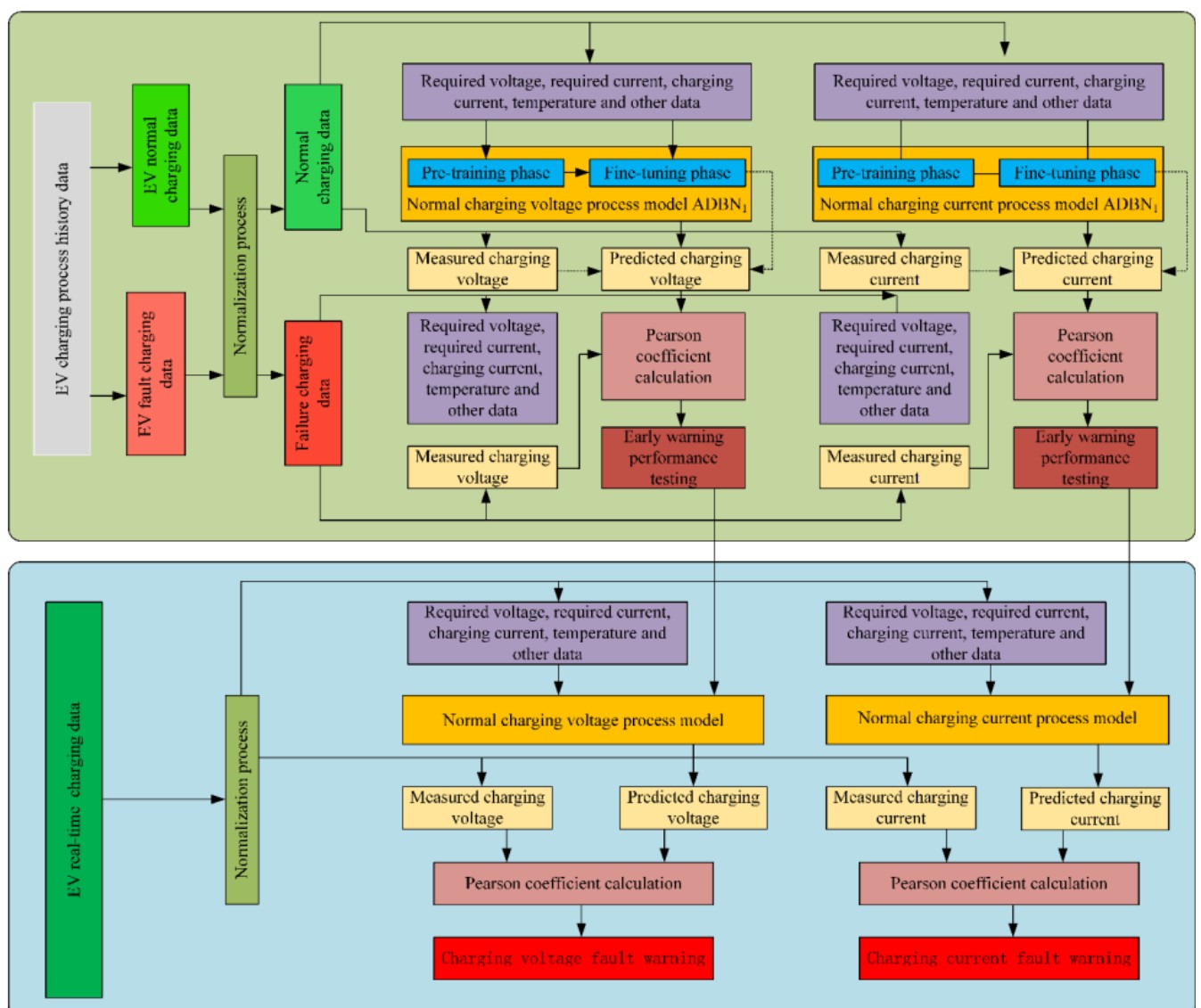

**Figure 3.** EV charging fault warning process.

## 4. Implementation of Fault Warning for EV Charging Process

### 4.1. Data Selection and Pre-Processing

Taking a Volkswagen Bora electric car with a nominal energy of power battery of 37.2 kWh as the experimental object, the historical charging data are collected and divided into normal charging data and fault charging data. In order to improve the accuracy of model prediction, 37,000 pieces of the normal charging data are selected for model training. By analyzing the update period of the physical quantities obtained in Table 1, the minimum update period is 50 ms. Considering the frequency of data update and fault warning detection, a fault detection is performed every 250 ms, i.e., the minimum update period is updated five times, which means a fault detection is optimal for every five elements of data. According to the above analysis, the fault charging data is sorted out and divided into charging voltage fault data set and charging current fault data set. Every five pieces of data in the fault charging data set are divided into one group, among which there are 2400 groups of charging voltage fault data and 2400 groups of charging current fault data.

In order to better reflect the data characteristics, improve the convergence speed of the ADBN model, and obtain higher fault warning accuracy, this paper uses the method of

extreme difference normalization to normalize the data set, and the calculation formula is as follows:

$$x_{out} = \frac{x_{in} - x_{min}}{x_{max} - x_{min}} \tag{16}$$

where $x_{max}$, $x_{min}$ are the maximum and minimum values of the same group of data in the input data respectively; $x_{out}$ is the normalized result of the input data $x_{in}$, which is used as the input of ADBN.

### 4.2. Training of ADBN Model

The training of the ADBN model was built using the Tensorflow library. The experimental software environment is Python 3.5.7 and tensorflow 1.2.1. The hardware configuration of the computer is Intel (R) core (TM) i5-3210M CPU @ 2.50GHZ and 8 GB of RAM.

By analyzing the physical quantities obtained in Table 1, the charging voltage and charging current are used as output quantities and the remaining variables are used as inputs, it is determined that the input layers of $ADBN_1$ and $ADBN_2$ are 15. Considering the accuracy of model prediction and training time, some network parameter settings of $ADBN_1$ and $ADBN_2$ are shown in Table 3.

**Table 3.** ADBN network parameters.

| Description | Symbol | Value |
|---|---|---|
| Maximum number of pre-training iterations | - | 50 |
| Fine-tune the maximum number of iterations | - | 50 |
| Number of pre-trained batch samples | - | 100 |
| Number of fine-tune batch samples | - | 100 |
| Learning rate of pre-training | - | 0.4 |
| Learning rate of fine-tuning | - | 0.001 |
| Exponential decay rate of first-order moments | $\beta_1$ | 0.9 |
| Exponential decay rate of second order moments | $\beta_2$ | 0.99 |
| Correction parameters | $\varepsilon$ | $10^{-8}$ |

In addition to determining the above parameters, it is also necessary to select the appropriate number of ADBN hidden layers $L$ and the number of hidden layer units $n$. Since the number of input and output variables and parameter values are the same for $ADBN_1$ and $ADBN_2$. Therefore, in this paper, we take $ADBN_1$ as an example to tests the influence of different number of hidden layers $L$ and different number of hidden layer units $n$ of $ADBN_1$ on the accuracy of charging voltage warning with the same parameters in Table 3. The number of hidden layers and the number of hidden layer units of $ADBN_2$ are also determined.

In order to study the influence of the number of hidden layers on the accuracy of charging voltage fault warning, the $ADBN_1$ networks with $L$ of 1, 3 and 5 are constructed. Set the number of units in the hidden layer of $ADBN_1$ to 100 and test with conducted with the same historical normal data and fault data. The accuracy of ADBN with different hidden layers is plotted with the number of iterations as a curve, as shown in Figure 4.

According to Figure 4, the accuracy of $ADBN_1$ is highest when $L = 3$, the lower accuracy when $L = 1$, and the lowest accuracy with a large change when $L = 5$. The reason for the large change of accuracy at $L = 5$ may be that the DBN model is more complex due to the increase of the number of hidden layers, which makes the optimization process of NAdam tortuous. According to the above analysis, the number of hidden layers of $ADBN_1$ is set to three, and then the influence of the number of units of different hidden layers on the accuracy is determined. Figure 5 is the change of accuracy corresponding to the number of units in different hidden layers.

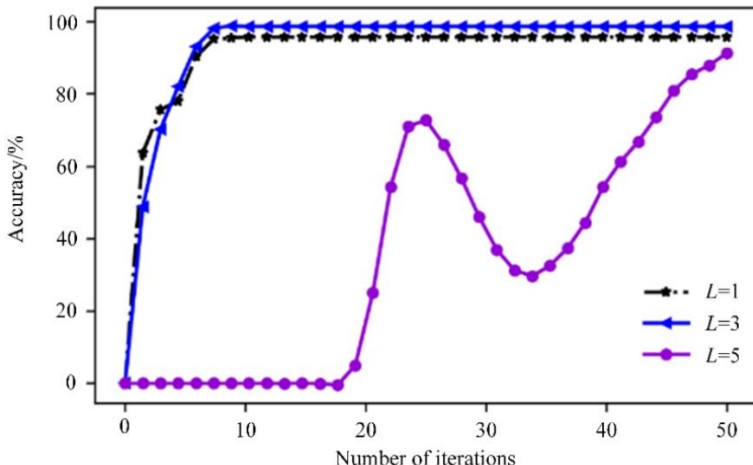

**Figure 4.** Change of accuracy corresponding to different number of hidden layers.

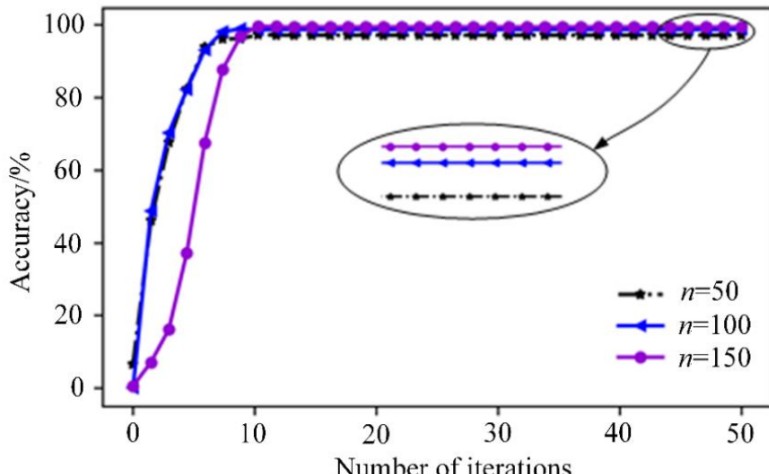

**Figure 5.** Change of accuracy corresponding to the number of units in different hidden layers.

As can be seen from Figure 5, when the number of hidden layer units $n$ in ADBN$_1$ is 50, 100 and 150, the warning accuracy is above 90%, and the warning accuracy increases with the increase of the number of hidden layer units. However, when $n$ is greater than 100, the improvement of accuracy is small. Considering the speed of model training and over-fitting problem, the network structure with the number of hidden layer units of 100 is selected in this paper.

Figure 6 shows the change curves of the ADBN$_1$ loss value and accuracy with the number of iterations in the fine-tuning stage. It can be seen from the observation curve that the loss error decreases rapidly from 0 to the 5th iteration, and gradually tends to a stable value after 10 iterations. Finally the loss error reaches 0.0087 at 50 iterations, and the average accuracy of voltage fault warning reaches 98.61% and stabilizes at this time.

According to the analysis results of Figure 4 to Figure 6, the structure of the voltage fault warning model for EV charging process is 15-100-100-100-1 (including input layer and BP output layer), that is, when the number of hidden layers is three and the number of hidden layer units is 100, it has the best warning effect.

When the structure of ADBN$_2$ is the same as ADBN$_1$, the accuracy of charging current fault warning can reach 97.75%, which proves that ADBN can be adapted to the corresponding charging data when predicting the physical quantities in different charging processes. Therefore, the structure of ADBN$_2$ can be the same as ADBN$_1$.

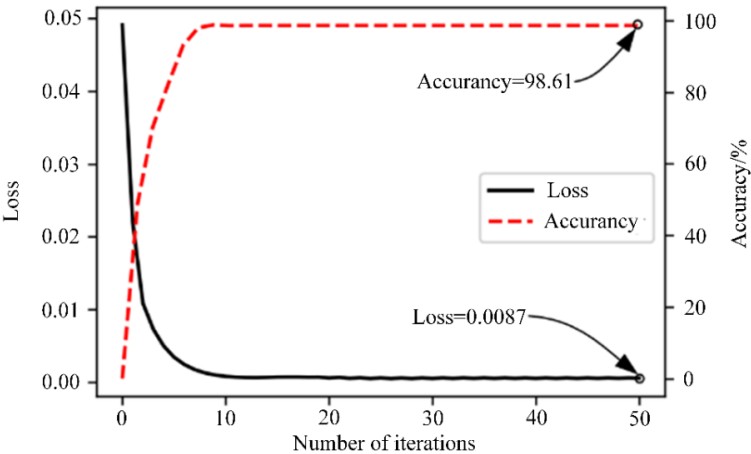

**Figure 6.** The change curve of ADBN$_1$ loss value and accuracy.

### 4.3. Method Validation and Comparison

In order to verify the effectiveness of the method in this paper, a group of charging data with charging voltage fault and a group of charging data with charging current fault of the Bolero EV car are randomly selected. The two groups of charging data are input into ADBN$_1$ and ADBN$_2$ to predict the output, and judge the Pearson coefficient with the actual measured output in order to analyze whether it can accurately warn the fault. The change of Pearson coefficients is shown in Figures 7 and 8.

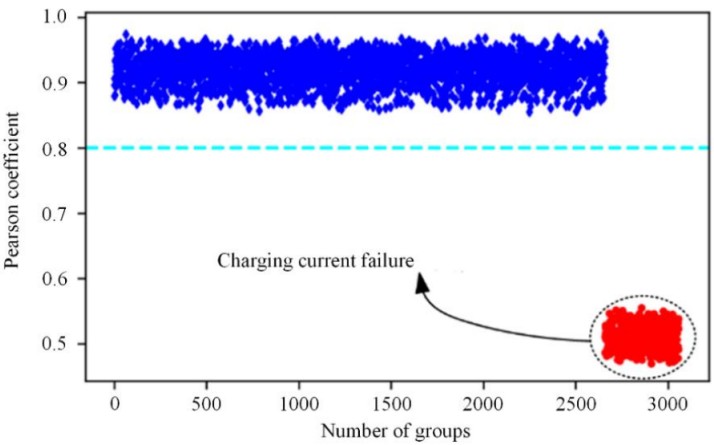

**Figure 7.** Change of Pearson coefficient of ADBN$_1$ predicted output and measured value.

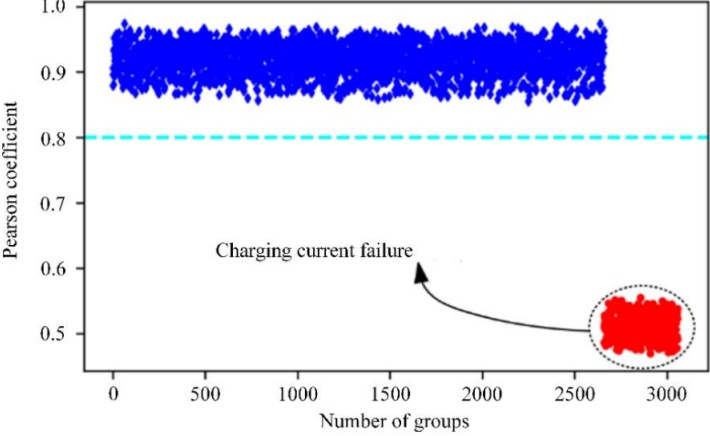

**Figure 8.** Change of Pearson coefficient of ADBN$_2$ predicted output and measured value.

As shown in Figures 7 and 8, when the EV is charged normally, the Pearson coefficient between the predicted value of the model and the actual measured value is in the range of 0.8 to 1. When a fault occurs, the Pearson coefficient is less than 0.8. This proves that the warning limit and the warning method selected in this paper can warn of a fault in the charging process well.

In order to verify the advancement of the method in this paper, it is compared and tested with the traditional DBN and the widely used BPNN. In the case of the same data set, the three are combined with the Pearson coefficient at the same time to compare the accuracy and the model convergence time. In order to ensure the accuracy of the experimental results, each test is repeated 10 times, and the average value of 10 tests is taken as the final warning evaluation index. The experimental results are shown in Table 4.

**Table 4.** Comparison of algorithm results.

| Methods | Model Alert Accuracy (%) | Model Convergence Time (s) |
|---|---|---|
| $ADBN_1$ | 98.61 | 424.16 |
| $DBN_1$ | 94.86 | 481.37 |
| $BPNN_1$ | 89.69 | 13.47 |
| $ADBN_2$ | 97.75 | 416.66 |
| $DBN_2$ | 94.62 | 487.94 |
| $BPNN_2$ | 90.65 | 13.97 |

In Table 4, $DBN_1$ and $DBN_2$ are DBN models for normal charging voltage and normal charging current of EV, and $BPNN_1$ and $BPNN_2$ are BPNN models for normal charging voltage and normal charging current of EV. The fine-tuning learning rate of DBN model is 0.6 and one momentum parameter is 0.25, and the rest of the network parameters and network structure are the same as ADBN. BPNN adopts a single hidden layer structure, the corresponding number of nodes is 15-9-1, the learning rate is 0.05, and the number of iterations is 1000.

The experimental results show that the accuracy of DBN has significantly improved compared with BPNN, which indicates that the deep network algorithm can realize deep data mining, has better processing ability for highly coupled and multidimensional data, and reflects better generalization ability. In addition, because the training process of DBN is divided into two stages of pre-training and fine-tuning, and the network is more complex, so the model convergence time of DBN is longer than that of BPNN. Due to the use of the NAdam algorithm, ADBN accelerates the gradient descent process and achieves dynamic adjustment of learning rate with strong adaptivity. Therefore, the accuracy of ADBN is higher than DBN and the model convergence time is shorter.

## 5. Conclusions

The method proposed uses NAdam as the optimization algorithm for DBN model training, which speeds up the parameter optimization process, shortens the training time of the model, has strong constraints on the learning rate, enhances the adaptability of the model, and can adaptively find the best parameters according to different input data. The method fully considers the characteristics of the multidimensionality, complexity and large amount of data of the EV charging process data, excavates the historical data of the EV charging process in depth, and solves the defects of the weak generalization ability and poor data feature extraction ability of traditional machine learning methods to a certain extent. Experiments prove that the proposed method has better warning effect in the application of fault warning of the EV charging process, and has higher warning accuracy than DBN and BPNN. The method proposed in this paper can only determine the charging voltage fault, charging current fault or other fault in an EV, but it cannot determine the specific type of fault type, such as charging voltage too high fault, charging current too high fault, etc. Methods should be added in the future to determine the specific types of faults that occur.

**Author Contributions:** Conceptualization, D.G. and Y.W.; methodology, D.G. and Y.W.; software, Y.W.; validation, D.G.; formal analysis, Y.W.; writing—original draft preparation, Y.W.; writing—review and editing, D.G., X.Z. and Q.Y.; visualization, Y.W.; All authors have read and agreed to the published version of the manuscript.

**Funding:** This research was funded by the National Natural Science Foundation of China (Grant No. 61673357), and the Natural Science Foundation of Shandong province of China (Grant No. ZR2018LF008), and the Key Research and Development Program of Shandong Province of China (Grant No. 2019GGX101012).

**Conflicts of Interest:** The authors declare no conflict of interest.

## Nomenclature

| | |
|---|---|
| EV | electric vehicle |
| BMS | battery management system |
| SOC | state of charge |
| ADBN | adaptive deep belief network |
| NAdam | Nesterov-accelerated adaptive moment estimation |
| DBN | deep belief network |
| BPNN | back propagation neural network |
| VMD | variational mode decomposition |
| UAV | unmanned aerial vehicle |
| RBM | restricted Boltzmann machine |
| Adam | adaptive moment estimation |

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
