# Peer review of "A Fault Warning Method for Electric Vehicle Charging Process Based on Adaptive Deep Belief Network"

_wevj, doi:10.3390/wevj12040265_

Round 1

Reviewer 1 Report

A fault warning method is proposed for EV charging process based on adaptive deep belief network(ADBN). The presented approach is somehow novel and shows some good results.However, there are still some places that need further revisions or clarifications. The detailed comments are listed below .

  1. In the introduction, it is suggested to supplement and analyze the references on fault warning method for EV charging.
  2. Some abbreviations, such as VMD and UAV, are not given full names in the introduction.
  3. This paper gives the ADBN structure of charging voltage and charging current faults, whether it is applicable to other types of fault warning?
  4. In Section 4 , a fault detection is performed every 250ms, but convergence time of ADBN1 is 424.16s. How to consider the real-time fault warning?
  5. The fine-tuning learning rate is 0.6 in DBN while it is 0.001 in ADBN. Does the obvious difference in parameter selection influence the conclusion?

Author Response

Response 1: We strongly agree with your suggestions. The revised version reviews other papers on charging faults, battery fault diagnosis and BMS equivalent circuit models, adds more literature sources, and adds and analyzes information on charging fault warning methods for electric vehicles. The revision is as follows:

Battery fault diagnosis methods can be divided into model-based methods and non-model-based methods[3, 4]. Zhang, et al, [5] a method for the monitoring and warning of EV charging faults based on a battery model is proposed to judge whether the charging process is normal by comparing the charging response information simulated by the battery model with the battery charging status information. The method is a model-based methods, which requires the establishment of an accurate electrochemical, electrothermal or other type of model depending on the type of battery. Tran, et al, [6] established the equivalent circuit model of lithium-ion battery that depends on the state of charge, temperature and state of health, which has a high accuracy and can be effectively monitored for lithium batteries. In [7], the performance of three different equivalent circuit models was studied and compared using the chemical composition of four kinds of lith-ium batteries, and the best model for each lithium battery was determined. Model-based methods are required signal processing and a large amount of calculation when dealing with complex and nonlinear systems, which makes them difficult to apply to a variety of EV, so the versatility is poor. The non-model-based method relies less on battery modeling, has stronger applicability, and has been widely used in the field of battery fault diagnosis. [8] proposes a fault diagnosis and identification method for EV based on BPNN, which can accurately identify charging fault levels and bring some assurance for EV charging safety.

You can see it in lines 46-63 of the revised version.

Response 2: Thank you very much for pointing out this error. The full names of VMD and UAV have been given in the revision and the acronyms and abbreviations used are summarized before the introduction. Details are as follows:

Nomenclature

EV : electric vehicle

BMS : battery management system

SOC : state of charge

ADBN : adaptive deep belief network

NAdam : Nesterov-accelerated adaptive moment estimation

DBN : deep belief network

BPNN : back propagation neural network

VMD : variational mode decomposition

UAV : unmanned aerial vehicle

RBM : restricted boltzmann machine

Adam : adaptive moment estimation

You can see it in lines 24-35 of the revised version.

Response 3: Thank you for your question, the ADBN structure given in this paper is applicable to other types of fault warnings. We actually considered other faults (temperature faults and SOC faults) in the EV charging process. Since the model training process and fault warning methods are the same for each fault, we only use the charging voltage fault and charging current fault as examples for experimental validation. Detailed explanation is as follows:

Charging voltage fault, charging current fault, temperature fault and SOC fault may occur mainly during the charging process of EV. Through Pearson correlation coefficient analysis, the fault types and fault discrimination methods are shown in Table 2. The fault warning of EV charging process can be mainly divided into two stages: EV normal charging model training and real-time fault warning. Because the model training method and fault warning method are the same for each fault, we only take the fault warning of charging voltage and charging current as an example to realize the charging fault warning of EV.

You can see it in lines 262-269 of the revised version.

Response 4: Thank you very much for your question. The fault warning method proposed in this paper is divided into two phases: normal charging model training and real-time fault warning, and the training of ADBN1 model belongs to the normal model training phase, while the fault detection belongs to the real-time fault warning phase, and the normal charging model training phase is before the real-time fault warning phase. Therefore, there is no direct connection between the convergence time of ADBN1 is 426.16 seconds and the fault detection is 250ms, that is, the convergence time of the model has no effect on the detection period.

Response 5: Thank you for your doubt, the obvious difference in parameter selection in DBN and ADBN does not influence the conclusions. Because ADBN uses the NAdam algorithm to take dynamic adjustment of learning rate in model training, and the NAdam algorithm suggests setting the learning rate to 0.001, so setting the learning rate of ADBN to 0.001 has better experimental results. However, DBN does not use the algorithm to optimize the learning rate in model training, and the experimental results are very poor when the learning rate is set to 0.001, so the learning rate of DBN cannot be set to 0.001. It is experimentally verified that DBN has the best experimental results when the learning rate is set to 0.6. Therefore, in this paper, we choose to set the fine-tuning learning rate of DBN to 0.6 and the fine-tuning learning rate of ADBN to 0.001.

We also corrected other minor errors and touched up the language of our manuscript, as detailed in the revised version.

Special thanks to you for your good comments.

Reviewer 2 Report

Paper must address following before acceptance:

(1) Battery power is mentioned, but charging voltage used in example is not mentioned. Tale several examples to verify your model for automobiles where battery charging is done at 400 V and 800 V.

(2) Charging cable can provide fault. Include in the model. This is true for high voltage and high power.

(3) Comment on the limitations of the model for heavy duty trucks where EV charger will operate almost at 1 MW. 

Author Response

Response 1: It is true as the reviewers pointed out that we only mentioned the battery power, but charging voltage used in example is not mentioned. Before answering this question, we need to clarify that the battery power mentioned in the article is only an explanation of the selected power battery parameters, and there is no direct connection between the nominal energy of the power battery and the charging voltage. In addition, the charging process data is collected when the electric vehicle is charged on a high-power DC charging pile. The high-power DC charging pile used in this paper has a rated output power of 150 kW and the voltage range of 400-750V for constant power charging. Therefore, our model for automobiles where battery charging is done at 400 V and 800 V.

Response 2: Thank you very much for your comment. The high power DC charging pile is shown in Figure 1.

As shown in Figures 1, the high-power DC charging pile is connected to the charging interface through the charging cable. The high-power DC charging pile has an output power rating of 150KW and the voltage range of 400-750V for constant power charging. When the charging interface is connected to the electric vehicle, the electric vehicle will send its own charging information and some internal fault information to the charging pile according to the communication protocol "GB/T 27930 Communication Protocol between Non-vehicle Conductive Charger and Battery Management System for Electric Vehicles".

Response 3: The DC voltage output range of the high-power DC charging pile is 200-750V, and the maximum output current is 250A. According to the power calculation formula P=UI, the charging power range of this charging pile is 50KW-187.5kW. The high-power charging pile selected in this paper can charge electric vehicle and electric physical vehicle, as well as large buses. Although it does not meet the charging demand of 1MW for heavy duty trucks, it still has strong applicability.

Reviewer 3 Report

Dear authors,

I appreciate your special efforts to carry out current scientific research in the field of electric vehicles. The presentation of this paper is of great clarity and high professionalism. The idea of ​​the EV charging process failure warning is well explained. The selection of the ADBN method to detect an EV charging process failure warning accurately is entirely justified by the results of a reasonable number of simulations. The ADBN superiority is demonstrated by comparison and testing with the traditional DBN and the widely used BPNN. Also, the impact of changing the number of hidden layers and the number of neurons in these layers is well-argued. The conclusions are supported by all these results obtained from the simulations. All that remains is to insert in the text a table with the acronyms and abbreviations used for better reading and understanding of the manuscript. Aso, please mention in the Conclusions section some of the research directions in your future work.

Author Response

Response 1: Thanks very much for your suggestion, the revised version has summarized the acronyms and abbreviations used before the introduction. Details are as follows:

Nomenclature

EV : electric vehicle

BMS : battery management system

SOC : state of charge

ADBN : adaptive deep belief network

NAdam : Nesterov-accelerated adaptive moment estimation

DBN : deep belief network

BPNN : back propagation neural network

VMD : variational mode decomposition

UAV : unmanned aerial vehicle

RBM : restricted boltzmann machine

Adam : adaptive moment estimation

You can see it in lines 24-35 of the revised version.

Response 2: We strongly agree with your comments, the research directions for future work have been described in the conclusion section of the revised version. The description is as follows:

The method proposed in this paper can only determine the charging voltage fault, charging current fault or other fault in EV, but it cannot determine the specific type of fault type, such as charging voltage too high fault, charging current too high fault, etc. Methods should be added in the future to determine the specific types of faults that occur.

You can see it in lines 425-428 of the revised version.

We also corrected other minor errors and touched up the language of our manuscript, as detailed in the revised version.

Special thanks to you for your good comments.

Reviewer 4 Report

This paper proposes a fault diagnosis method during the charging process of electric vehicles (EVs). The method is based on an adaptive deep belief network, using Nesterov-accelerated adaptive moment estimation to optimize the training process and using historical EV charging data to build the normal charging process of EVs. The research topic and results from this paper are interesting but a bit too simple, so before it can be considered for publication, some issues should be addressed first.

  1. The authors need to double-check the English grammar, figure quality, and formatting throughout the paper, in order to make it more legible for readers. For example, font and font size are inconsistent throughout the paper; it should be “Figure #. Text” not “Figure .# Text”; the quality of all the figures needs to be improved, especially Figures 1, 2, and 3, since they are not legible at the moment; lines 80-84, complete sentences cannot be started with verbs as such; etc.
  2. The literature review of this paper is quite weak. The authors need to add more literature sources to strengthen the background of the paper.
  • Reviewing other papers on the same topic of charging fault: https://doi.org/10.1088/1742-6596/1646/1/012046, https://doi.org/10.3390/wevj12010014.
  • Reviewing battery fault diagnosis: https://doi.org/10.3390/a13030062, https://doi.org/10.1155/2015/631263.
  • Reviewing the BMS and comprehensive equivalent circuit models used in the BMS: https://doi.org/10.1016/j.est.2021.103252, https://doi.org/10.3390/batteries7030051.
  1. What are the faults that can happen during the EV charging process? What would be the indicators of those faults (based on measurements of voltage, current, temperature, or other derived metrics)? The authors should review/provide this information in the form of a table. Even though the faults are being detected using machine learning, it is still important for the algorithm developers to understand the mechanism of the faults to avoid false diagnoses, since historical data can only provide a small picture of the science behind the inner workings of Li-ion batteries and EVs.
  2. The authors use charging voltage fault and charging current fault data to validate their algorithms.
  • There are probably more types of faults during EV charging than these two (temperature, SOC, etc.). Did the authors consider those?
  • What are the characteristics of these voltage and current faults (bias/offset, gain/scaling, etc.)? Were they simulated or did they come from actual real-life data? If they were simulated, then the authors should consider adding more types of faults to validate their algorithms.
  1. It is understood that the fault diagnosis algorithms proposed are used for fault alarming. However, what is the capability of the proposed approach in terms of fault isolation (identifying exactly what type of faults occurs)? The authors should include and discuss this with the findings. This would significantly contribute to the practicality of the proposed method.

Author Response

Response 1: Thank you very much for pointing out this error. The revised version has been double-check for English grammar, figure quality and formatting. First, the revised version has improved the quality of all figures and changed all "Figure . # Text" to "Figure #. Text". Then, the inconsistent font of the secondary headings in the revised version was changed, and the font size and formatting in the table headings were also modified. Finally, lines 80-84 were modified, and the details of the changes are shown as follows:

The real-time physical quantities of EV charging are obtained and input into the con-structed exclusive deep network model to get the desired prediction data. The correlation between the predicted data and the actual measurements is observed to determine whether there is a fault in EV charging, thus realizing fault warning for the EV charging process.

You can see it in lines 127-131 of the revised version.

Response 2: Thank you very much for your comments. The revised version reviews other papers on charging faults, battery fault diagnosis and BMS equivalent circuit models, and more literature sources have been added. The revision is as follows:

Battery fault diagnosis methods can be divided into model-based methods and non-model-based methods[3, 4]. Zhang, et al, [5] a method for the monitoring and warning of EV charging faults based on a battery model is proposed to judge whether the charging process is normal by comparing the charging response information simulated by the battery model with the battery charging status information. The method is a model-based methods, which requires the establishment of an accurate electrochemical, electrothermal or other type of model depending on the type of battery. Tran, et al, [6] established the equivalent circuit model of lithium-ion battery that depends on the state of charge, temperature and state of health, which has a high accuracy and can be effectively monitored for lithium batteries. In [7], the performance of three different equivalent circuit models was studied and compared using the chemical composition of four kinds of lith-ium batteries, and the best model for each lithium battery was determined. Model-based methods are required signal processing and a large amount of calculation when dealing with complex and nonlinear systems, which makes them difficult to apply to a variety of EV, so the versatility is poor. The non-model-based method relies less on battery modeling, has stronger applicability, and has been widely used in the field of battery fault diagnosis. [8] proposes a fault diagnosis and identification method for EV based on BPNN, which can accurately identify charging fault levels and bring some assurance for EV charging safety.

You can see it in lines 46-63 of the revised version.

Response 3: Thank you very much for your comments. The revised version has described the possible faults in the charging process of EV and given the possible faults in the charging process of EV and their identification methods in the form of a table. The table of fault types and identification methods is shown in Table 1.

Table 1. Fault type and identification method table

Number

Fault type

Fault identification method

1

Charging voltage fault

The Pearson coefficient between model prediction and actual measurement is less than 0.8

2

Charging current fault

The Pearson coefficient between model prediction and actual measurement is less than 0.8

3

Temperature fault

The Pearson coefficient between model prediction and actual measurement is less than 0.8

4

SOC fault

The Pearson coefficient between model prediction and actual measurement is less than 0.8

You can see it in lines 262-264 and 287 of the revised version.

Response 4:

(1) Thank you for your question, we actually considered other faults (temperature faults and SOC faults) in the EV charging process. Since the model training process and fault warning methods are the same for each fault, we only use the charging voltage fault and charging current fault as examples for experimental validation. Detailed explanation is as follows:

Charging voltage fault, charging current fault, temperature fault and SOC fault may occur mainly during the charging process of EV. Through Pearson correlation coefficient analysis, the fault types and fault discrimination methods are shown in Table 2. The fault warning of EV charging process can be mainly divided into two stages: EV normal charging model training and real-time fault warning. Because the model training method and fault warning method are the same for each fault, we only take the fault warning of charging voltage and charging current as an example to realize the charging fault warning of EV.

You can see it in lines 262-269 of the revised version.

(2) Thank you for your doubt, the charging voltage fault and charging current fault mentioned in this article are characterized by bias/offset. The fault occurs when the charging voltage and charging current are higher or lower than the required voltage and required current at the time of charging. The fault data used in this article is from actual real data.

Response 5: Thank you for your comments. The method proposed in this paper can only identify the charging voltage fault, charging current fault or other fault in EV, but not the specific fault types, such as charging voltage too high fault, charging current too high fault, etc. How to identify the specific fault type is the focus of our research afterwards. The revised version discusses this in the study results, and the specific description is as follows:

The method proposed in this paper can only determine the charging voltage fault, charging current fault or other fault in EV, but it cannot determine the specific type of fault type, such as charging voltage too high fault, charging current too high fault, etc. Methods should be added in the future to determine the specific types of faults that occur.

You can see it in lines 425-428 of the revised version.

We also corrected other minor errors and touched up the language of our manuscript, as detailed in the revised version.

Special thanks to you for your good comments.

Round 2

Reviewer 1 Report

In line 96“If the Pearson coefficient exceeds the set threshold, the fault will be warned ”,here “exceed “generally means “be greater than”. But in table 2” The Pearson coefficient between model prediction and actual measurement is less than 0.8”.

Author Response

Response 1: Thank you very much for pointing out this error. We have replaced all “exceed” with “less than” in the revised version. For example, “If the Pearson coefficient exceeds the set threshold” is modified to “If the Pearson coefficient less than the set threshold” and “perform fault warning when the Pearson coefficient exceeds the set expectation value” is modified to “perform fault warning when the Pearson coefficient less than the set expectation value”.

You can see in the revision in lines 96 and 254.

We also corrected other minor errors and touched up the language of our manuscript, as detailed in the revised version.

Special thanks to you for your good comments.

Reviewer 4 Report

The authors have improved the manuscript based on the reviewer's comments. However, there are still a couple of points to be addressed before this manuscript can be accepted for publication:

  • Ref [6] is incorrect; it should be from https://doi.org/10.1016/j.est.2021.103252
  • Table 2 should have another column after Fault type, being Fault description (describe the faults - bias fault and what that means).

Author Response

Response 1: Thank you very much for pointing out this error. We have revised the ref [6] in the revised version. The modifications are shown as follows:

Tran, M.K.; Mathew, M.; Janhunen, S.; Panchal, S.; Raahemifar, K.; Fraser, R.; Fowler, M. A comprehensive equivalent circuit model for lithium-ion batteries, incorporating the effects of state of health, state of charge, and temperature on model parameters. Journal of Energy Storage 2021, 43, 103252.

You can see it in lines 415-417 of the revised version.

Response 2: Thank you very much for your comments. The revised version adds a fault description column to Table 2 to describe the characteristics of the fault and its meanings. The revision is shown as follows:

Table 2. Fault type and identification method table

Number

Fault type

Fault identification method

Fault description

1

Charging voltage fault

The Pearson coefficient between model prediction and actual measurement is less than 0.8

Bias fault - charging voltage is higher or lower than the required voltage

2

Charging current fault

The Pearson coefficient between model prediction and actual measurement is less than 0.8

Bias fault - charging current is higher or lower than the required current

3

Temperature fault

The Pearson coefficient between model prediction and actual measurement is less than 0.8

Bias fault - the measured value of temperature is widely different from the predicted value

4

SOC fault

The Pearson coefficient between model prediction and actual measurement is less than 0.8

Bias fault - the measured value of SOC is widely different from the predicted value

You can see it in lines 259 of the revised version.

We also corrected other minor errors and touched up the language of our manuscript, as detailed in the revised version.

Special thanks to you for your good comments.
